# Genomic and chemical insights into a human lectin-binding extracellular polysaccharides from *Parageobacillus toebii* strain H-70

Diana Ghevondyan[1,2], Armine Margaryan [1,2], Fabrizio Chiodo[3,4], Yvette van Kooyk[4], Ilaria Finore[3]*, Annarita Poli[3], Hovik Panosyan [2,5]*

1 Department of Biochemistry, Microbiology and Biotechnology, Yerevan State University, Yerevan, Armenia, 2 Research Institute of Biology, Biology Faculty, Yerevan State University, Yerevan, Armenia, 3 Institute of Biomolecular Chemistry, National Council of Research (C.N.R.), Pozzuoli, Naples, Italy, 4 Department of Molecular Cell Biology and Immunology, Amsterdam UMC, Vrije Universiteit Amsterdam, Amsterdam, The Netherlands, 5 Department of Biomedical Sciences, Institute of Pharmacy, Yerevan State University, Yerevan, Armenia

* hpanosyan@ysu.am (HP); ilaria.finore@cnr.it (IF)

## Abstract

Extracellular polysaccharides (EPSs) from thermophilic bacteria are promising biopolymers due to their stability and structural variability. This study aimed to characterize the genomic and chemical features of EPS produced by *Parageobacillus toebii* strain H-70 isolated from a geothermal spring in Armenia. EPS from strain H-70 was produced in sucrose and glucose based medium and analyzed chemically by TLC, HPAEC-PAD, GC-MS, and NMR. Protein, uronic acid, and nucleic acid contents were quantified by spectrophotometric methods. Molasses, an inexpensive byproduct of sugar production, was used as the carbon source too. Whole-genome sequencing, comparative phylogenomics, and genome mining were performed to identify biosynthetic gene clusters, carbohydrate-active enzymes (CAZymes), and regulatory components associated with EPS metabolism. Strain H-70 yielded 37.9 mg/L EPS (0.10 g/g dry cell weight) after 72 h cultivation at 55 °C and pH 7.0 with sucrose as sole carbon source. The EPS was a heteropolysaccharide composed of rhamnose, glucose, galactose, and mannose, along with proteins (15.04%), uronic acids (4.22%), and nucleic acids (4.88%). The EPS yield obtained with glucose as the sole carbon source was 10.5 mg/L, whereas molasses supplementation resulted in a yield of 14.5 mg/L. The draft genome (~3.2 Mb, 42% G + C, 98.9% ANI with *P. toebii* DSM 14590) encoded five Wzy-dependent EPS gene clusters with glycosyltransferases, transporters, and regulators. The genome also carried diverse CAZymes (GH, GT, CE, CBM, AA families) and modification enzymes (e.g., CsaB, acetyltransferases), indicating structural and functional variability of the polymer. In addition, the binding to human C-type lectins (carbohydrate-binding proteins involved in innate and adaptive immune-responses) has been studied by solid-phase assay. This study provides

**Data availability statement:** All relevant data are within the manuscript and its Supporting information files.

**Funding:** This research was supported by the Higher Education and Science Committee of MESCS RA (Research projects N 23SC-CNR-1F010 and 23AA-1F043), by the European Commission – NextGenerationEU Project MICS (Made in Italy – Circular and Sustainable) n. PE00000004, CN_00000033 "National Biodiversity Future Center"- CUP B83C22002930006, PRIN MUR PNRR 2022 (P202293ZMC), FutuRaw FOE 2022 DCM. AD005.081, and MIMIT SANATEC F/350309/03/X60. The funders had no role in study design, data collection and analysis, decision to publish, or preparation of the manuscript.

**Competing interests:** The authors have declared that no competing interests exist.

the first comprehensive characterization of EPS from *P. toebii* H-70, integrating genomic and chemical insights. The binding to human C-type lectins offers future EPSs biomedical applications especially in as immune-modulators.

## Introduction

Many microorganisms synthesize high-molecular-weight extracellular polysaccharides (EPSs), which form a major component of their cell envelope [1–4]. EPSs fulfill diverse ecological and physiological roles: they protect cells against environmental stress, desiccation, protozoan grazing, and viral attack, while also promoting surface adhesion and biofilm formation [5,6].

EPSs are broadly classified into homopolysaccharides, composed of a single sugar type, and heteropolysaccharides, which contain multiple monosaccharides along with organic (e.g., lactate, acetate, glycerol) and inorganic (e.g., phosphate, metal ions) substituents [7,8]. Their composition depends on the producing microorganism, nutrient availability, and environmental conditions [9]. Owing to unique physicochemical and rheological properties, microbial EPSs are increasingly exploited in food, pharmaceutical, paint, wastewater, and bioremediation industries [7,10].

EPS-producing microbes inhabit diverse ecosystems, including extreme environments such as hot springs, polar regions, and hypersaline habitats [2,11–15]. In extremophiles, EPSs provide critical adaptations to harsh conditions.

Thermophilic EPSs are particularly attractive for biotechnology because of their inherent thermal stability and functional robustness. Geothermal springs are rich sources of EPS-producing thermophiles, especially bacilli, which are metabolically versatile and capable of utilizing a wide range of substrates. EPS-producing thermophilic bacilli, including members of the genera *Bacillus*, *Geobacillus*, *Parageobacillus*, *Anoxybacillus*, *Aeribacillus*, and *Brevibacillus*, have been isolated from geothermal springs in Italy, Bulgaria, China, Turkey, Antarctica, and Armenia [2,8,16–25]. However, systematic studies of EPSs in thermophilic bacilli remain limited, despite their promising industrial potential [8,16,20,21,24]. Commercialization of EPS-based products is still in its early stages and demands further investigation to improve production efficiency. The high cost of EPS synthesis remains a major barrier. Adopting a circular bioeconomy approach could address this challenge by converting low-cost by-products, such as molasses and whey, into valuable substrates for EPS production, particularly using extremophilic microorganisms [26]. In bacteria, EPS biosynthesis follows conserved steps: activation of monosaccharides into nucleotide diphosphate sugars (NDP-sugars), assembly of repeating units by glycosyltransferases, polymerization at the membrane, and export to the cell surface [6,27]. Three major pathways are known: (1) Wzx/Wzy-dependent pathway, responsible for complex heteropolysaccharides; (2) ABC transporter-dependent pathway, mainly linked to capsular polysaccharides but also EPS; and (3) synthase-dependent pathway, which produces homopolysaccharides such as curdlan [28,29].

Genes for EPS biosynthesis are usually organized in clusters, though arrangements vary across taxa. For instance, *Lactobacillus* species possess a conserved epsABCDE core, whereas cyanobacteria often have scattered or multiple gene copies [27,30]. Regulation is equally complex: the AbrB protein acts as a global repressor, while MarR and LysR family regulators activate EPS gene expression [6].

Whole-genome sequencing has become essential for elucidating EPS biosynthetic pathways and linking gene content to polymer structure and function [6,31]. In this study, we investigated the EPS produced by *Parageobacillus toebii* H-70, a thermophilic isolate from a high-altitude geothermal spring in Hankavan, Armenia. Genomic analysis was performed to identify candidate genes involved in EPS biosynthesis, assembly, secretion, and regulation, complemented by chemical characterization of the polymer.

The recognition of carbohydrate-patterns is a fundamental immunological mechanism that facilitates communication between microbes and their hosts [32]. This process is primarily mediated by pattern recognition receptors, most notably the conserved family of carbohydrate-binding proteins known as lectins. While the interactions between human C-type lectins and pathogenic microbial glycans are well-documented, the immunomodulatory potential of EPS produced by thermophilic bacteria is poorly documented. To address this knowledge gap, the binding of the EPS produced by *P. toebii* H-70 with key human C-type lectins has been also studied.

## Materials and methods

### Screening of EPS-producing strains

More than 100 thermophilic isolates belonging to the genera *Parageobacillus*, *Brevibacillus*, *Bacillus*, *Geobacillus*, *Ureibacillus*, and *Anoxybacillus* were screened for EPS production. All strains, originally isolated from Armenian geothermal springs and preserved at Yerevan State University, were previously identified to the species level by phenotypic analysis and 16S rRNA gene sequencing [24]. Field sampling and isolation of microbial strains from geothermal springs were conducted within the framework of a state-funded research project officially approved by the Ministry of Education, Science, Culture and Sports of the Republic of Armenia, which authorized access to the field sites and permitted the collection of environmental samples. No additional individual permits were required.

Screening was carried out at 50–65 °C and pH 7.0–7.5 on minimal medium A (g/L: carbon source, 6.0; yeast extract, 0.2; peptone, 0.1; $MgSO_4$ x $7H_2O$, 0.1; KCl, 0.2) solidified with agar. Glucose, fructose, sucrose, arabinose, ribose, xylose, and galactose were tested individually as sole carbon sources. Colonies with mucoid morphology were considered EPS producers. Since *P. toebii* H-70 showed the highest production, was selected for deeper investigation.

### Biomass production and EPS recovery

*P. toebii* H-70 was pre-cultivated for 24 h in Nutrient Broth (Liofilchem, Italy), then transferred to buffered ($Na_2HPO_4$ x $7H_2O$, 16.35 g/L and $NaH_2PO_4$, 5.4 g/L) minimal medium A supplemented with sucrose or glucose 2% (w/v). Moreover, molasses served as an economical substrate for microbial growth and EPS production. The molasses, provided by Co. Pro.B (Cooperativa Produttori Bieticoli, Italy; http://www.coprob.com), contained 50% (w/v) sucrose [4] and was incorporated into buffered minimal medium A. Cultures were grown in flasks at 55 °C, 150 rpm, and pH 7.5 and checked up to 96 h. Growth was monitored by $OD_{540}$ measurements (Thermo Scientific GENESYS UV-Vis). Biomass and EPS yields were evaluated at exponential, early stationary, and late stationary phases. The EPS specific yield was calculated based on 1 g of dry cell weight (DCW). For biomass determination, 150 mL samples were centrifuged (7,000 rpm, 20 min, 4 °C), and pellets were freeze-dried and weighed by analytical balance. EPS was precipitated from culture supernatants with two volumes of cold absolute ethanol, kept at −20 °C overnight, and centrifuged (10,000 rpm, 45 min, 4 °C). Pellets were dissolved in warm distilled water (70 °C), dialyzed (12–14 kDa cut-off, 3–5 days) against tap water, lyophilized using a Heto dry winner and weighed.

## Chemical characterization of EPS

The carbohydrate content of the crude exoproduct (EP) was quantified using the phenol–sulfuric acid method, with glucose as the standard [33]. A standard calibration curve was prepared using glucose at a concentration of 1 mg/mL. Protein content was determined by the Lowry method [34], employing bovine serum albumin (BSA) as the standard across a concentration range of 0–50 µg/mL. Uronic acid concentration was measured using a colorimetric assay with *m*-hydroxybiphenyl reagent and glucuronic acid as the standard [35]. Nucleic acid content was assessed spectrophotometrically at λ260 nm, applying a conversion factor of 50 for double-stranded DNA. All measurements were performed in duplicate.

## EPS chemical characterization

For monosaccharide composition analysis, EPS was hydrolyzed with 0.5 M trifluoroacetic acid at 120 °C for 2 h. Hydrolysates were analyzed by thin-layer chromatography (TLC) and high-pressure anion-exchange chromatography with pulsed amperometric detection (HPAEC-PAD). TLC was performed on silica plates with *n*-butanol/acetic acid/water (6:2:2, v/v/v) and visualized using α-naphthol reagents [36]. HPAEC-PAD analyses were performed on a Dionex ICS-5000 + DC system with a CarboPac PA1 column and 16 mM NaOH eluent at a flow rate of 0.25 mL/min. Thermogravimetrical analysis (TGA) of EPS was performed with TGA/DTA Perkin-Elmer PyrisDiamond, with gas station. Approximately 4 mg of sample (kept under vacuum at 60 °C for 24 h prior analysis) were placed in a ceramic crucible and heated from room temperature up to 600 °C at a speed rate of 10 °C/min, under nitrogen flux at 30 mL/min.

## Gas chromatography–mass spectrometry (GC-MS) analysis

Monosaccharide composition was also determined by **Gas chromatography–mass spectrometry (**GC–MS) after derivatization to acetylated methyl glycosides [37]. EPS (0.5 mg) was methanolyzed (1.25 M HCl in methanol, 80 °C, 16 h), acetylated (acetic anhydride/pyridine, 100 °C, 30 min), and analyzed on a Thermo Scientific Focus GC with a TG-SQC column (30 m × 0.25 mm × 0.25 µm). Helium was the carrier gas (1 mL/min). The oven was programmed from 150 °C (3 min hold) to 330 °C at 3 °C/min.

[1]H Nuclear magnetic resonance (NMR) spectrum was recorded on a Bruker 400 MHz spectrometer at 50 °C. Sample were exchanges twice with $D_2O$, lyophilized and dissolved in 700 µL of $D_2O$. Chemical shifts (δ) were reported in parts per million (ppm), referenced to sodium 2,2,3,3-d[4]-(trimethylsilyl)propanoate for [1]H NMR [4].

## Genome sequencing and comparative genomics

The genomic DNA was extracted using a GenElute Bacterial Genomic DNA Kit (Sigma-Aldrich) and sequenced using Illumina paired-end technology (Eurofins Genomics, Germany, www.gatc-biotech.com). Reads were assembled with CLC Genome Workbench.

Digital DNA–DNA hybridization (dDDH) was calculated using Genome-to-Genome Distance Calculator GGDC 2.1 [38] against reference genomes (*Parageobacillus toebii* DSM 14590[T], *Parageobacillus galactosidasius* DSM 18751[T], *Parageobacillus yumthangensis* MTCC 12749[T], *Aeribacillus pallidus* KCTC 3564[T], *Saccharococcus thermophilus* DSM 4749[T], *Parageobacillus thermantarcticus* DSM 9572[T], *Parageobacillus caldoxylosilyticus* DSM 12041[T], *Parageobacillus thermoglucosidasius* DSM 2542[T], *Anoxybacillus karvacharensis* DSM 106524[T], *Anoxybacillus flavithermus* DSM 2641[T], *Geobacillus thermodenitrificans* DSM 465[T], *Anoxybacteroides tepidamans* DSM 16325[T] and *Geobacillus subterraneus* KCTC 3922[T]) (http://ggdc.Dsmz.De/distcalc2.Php). Phylogenetic trees were generated with TYGS (https://tygs.dsmz.de/) [38]. Average nucleotide identity (ANI) was determined using OrthoANIu [39,40] via EzBioCloud (https://www.ezbiocloud.Net/tools/orthoaniu).

## C-type lectins binding by ELISA-based solid-phase assay

Nunc MaxiSorp wells were coated overnight at 4 °C with 50 µL of EPS solution (10 µg/mL in 10 mM PBS, pH 7.4). Following coating, the wells were washed twice with 150 µL of TSM buffer (20 mM Tris-HCl, 150 mM NaCl, 1 mM CaCl$_2$, 2 mM MgCl$_2$, pH 8.0). Non-specific binding sites were subsequently blocked with 80 µL of 1% bovine serum albumin (BSA; Sigma-Aldrich, ≥ 96%) in TSM for 30 min at room temperature. After discarding the blocking solutions, 50 µL of human-Fc fused C-type lectins (DC-SIGN, Langerin, MGL, and MR) at a concentration of 1 µg/mL were added to the wells and incubated for 1 h at room temperature. Post-incubation, unbound lectins were removed by washing (twice with 150 µL of TSM buffer), and 100 µL of horseradish peroxidase (HRP)-conjugated goat anti-human IgG antibody (JacksonImmuno) at 0.3 µg/mL were added for a 30 min incubation. Following a final wash (twice with 150 µL of TSM buffer), 100 µL of 3,3′,5,5′-tetramethylbenzidine (TMB) substrate solution was added. The colorimetric reaction was allowed to proceed for 5 min before being stopped with 50 µL of 0.8 M H$_2$SO$_4$. Absorbance was measured at 450 nm using an ELISA plate reader. The experiments were performed in duplicate. Absorbance values were normalized to the signal generated by the positive controls for each lectin. The following glycan-functionalized polyacrylamide polymers (Lectinity MW approx. 20 kDa, carbohydrate content around 20% mol) were used as positive controls at 20 µg/mL: PAA-LeX (for DC-SIGN), PAA-Tn (for MGL), and PAA-LeY (for Langerin). Mannans from *Saccharomyces cerevisiae* (Sigma-Aldrich) were used at 10 µg/mL as a positive control for MR.

## Genome annotation and EPS gene cluster analysis

Annotation was performed using National Center for Biotechnology Information (NCBI) Prokaryotic Genome Annotation Pipeline (PGAP) [41] (https://www.ncbi.nlm.nih.gov/genome/annotation_prok/), Rapid Annotation using Subsystem Technology (RAST) [42] (http://rast.nmpdr.org/), the Department of Energy Systems Biology Knowledgebase (KBase) (https://www.kbase.us/), UniProt [43] (https://www.uniprot.org/), and the Kyoto Encyclopedia of Genes and Genomes (KEGG) [44] (https://www.kegg.jp/kegg/). Carbohydrate-active enzymes (CAZy) were identified with dbCAN2 HMMs [45]. EPS biosynthetic clusters were predicted with antiSMASH v7.0 [46] and visualized using Proksee (https://proksee.ca/) [47].

Comparative analysis of EPS clusters between H-70, *P. thermoglucosidasius* DSM 2542[T], *P. galactosidasius* DSM 18751[T], and *P. thermantarcticus* DSM 9572[T] was performed using BLASTp within pyGenomeViz (https://moshi4.github.io/pyGenomeViz/) [48]. Functional annotation of sugar metabolic pathways was refined using KAAS (https://www.genome.jp/kegg/kaas/) [49–51] and manual BLASTp validation. Comparative CAZy family distributions were visualized as stacked bar charts in Python 3.12.2.

## Statistical analysis

Data is expressed as mean ± standard error (SEM). Statistical significance was assessed using paired t-tests, with $p < 0.05$ considered significant. In C-type lectins binding by ELISA-based solid-phase assay, 2way ANOVA multiple comparison was performed with Dunnett's multiple comparisons test (alpha = 0.05) using GraphPad Prism 10.6.1. ns: $p > 0.05$, **** $p ≤ 0.0001$.

## Nucleotide sequence accession number

The draft genome sequence of *P. toebii* H-70 is available in GenBank under accession number JAZHOX000000000.

## Ethical statement

This article does not contain any studies with human participants or animals performed by any of the authors.

## Results

### EPS production and chemical characterization

From over 100 thermophilic bacilli isolated from Armenian geothermal springs, *P. toebii* strain H-70 was selected for its strong EPS-producing phenotype, forming mucous-like colonies on minimal medium A enriched by glucose and sucrose

(S1 Fig in S1 File). The strain has been deposited in the Microbial Depository Center of Armenia (MDC 11862). EPS and biomass production were monitored at 55 °C, 150 rpm, for 72–96 h. Biomass peaked during the stationary phase (48–72 h), with EPS production closely correlated (S2 Fig in S1 File). EPS synthesis began in early exponential phase and reached a maximum at 72 h, yielding 37.9 mg/L (0.10 g/g dry cell weight) for sucrose and 10.5 mg/L in case of glucose. When molasses was used as a sole carbon source EPS yield was 14.5 mg/L (S3 Fig in S1 File).

EPS samples were obtained after 72 h of cultivation at 55 °C, pH 7.0, and 150 rpm in sucrose containing medium and subjected to chemical and structural analyses. Based on TLC (S4 Fig in S1 File), HPAE-PAD, and GC-MS results, the EPS produced by the H-70 strain was identified as a heteropolysaccharide composed of rhamnose, glucose, galactose, and mannose (Fig 1). The extracellular fraction at 72 h contained proteins (15.04%), nucleic acids (4.88%), and uronic acids (4.22%). Determination of the thermal behavior of EPS is important in order to evaluate its commercial application in packaging or other industrial use. Thermogravimetric analysis is a thermal analysis in which the mass of the sample is measured over time as the temperature changes. It can be used to evaluate the thermal stability of the material. S5 Fig in S1 File shows TGA of EPS. Firstly, a step around 80–100 °C was observed that could be related to water evaporation from the EPS. This behavior is due to the initial loss of moisture related to the presence of high content of carboxyl groups, which are bound to water molecules [4]. A second step, related to the degradation of bigger part of the sample, was centered between 200 and 350 °C (at 300 °C, more than 50% of EPS was still present). According to thermal characteristics of EPS which are comparable to those from commercial biopolymers, this polymer could be considered as thermostable and could be therefore used in thermal processed below 200 °C without significant thermal degradation risk.

## Genome features and phylogeny

The draft genome of strain H-70 is ~3.2 Mb across 71 contigs, with 99× coverage, 42.0 mol% G + C content, 3509 CDSs, 65 RNA genes, and one CRISPR array (Table 1). Functional classification (S6 Fig in S1 File) showed enrichment in carbohydrate metabolism (224 CDSs), amino acid metabolism (268 CDSs), protein metabolism (164 CDSs), RNA metabolism (52 CDSs), secondary metabolism (9 CDSs), membrane transport (34 CDSs), stress response 20 CDSs and cell wall/capsule functions (28 CDSs). Comparative *in-silico* GGDC analyses revealed 92.3% genome similarity with *P. toebii* DSM 14590$^T$ and an ANI value of 98.91%, confirming assignment to *P. toebii*. dDDH values were calculated by comparing the genome of strain H-70 with the genomes of 14 different strains from genera *Aeribacillus, Saccharococcus*, *Parageobacillus*, *Anoxybacillus* and *Geobacillus* (S1 Table in S1 File). A phylogenomic tree (Fig 2) grouped H-70 with *P. toebii* DSM 14590$^T$, *P. galactosidasius* DSM 18751$^T$, and *P. yumthangensis* MTCC 12749$^T$.

## EPS genes clusters

AntiSMASH analysis identified five chromosomal EPS clusters encoding Wzy-dependent pathway components, including precursor biosynthesis enzymes, glycosyltransferases, transporters, and regulators (Fig 3). Thus, Cluster 1 (~26 kb) encodes maltodextrin ABC transporters, likely for sugar uptake; Cluster 2 (~34 kb) includes EamA and RND transporters, CsaB transferase, multiple GTs, LuxR regulators, and hypothetical proteins linked to EPS biosynthesis; Cluster 3 (~16 kb) contains sugar transporters, precursor enzymes, and mobile elements; Cluster 4 (~30 kb) encodes regulatory/structural proteins for EPS polymerization and export, including EpsC, EpsD, Wzb, Wze, and Wzx (peg.1503, 100% identity to *P. toebii*). It can be subdivided into five functional segments; Cluster 5 (~24 kb) harbors precursor enzymes and hypothetical proteins, including peg.1601, identical to *P. toebii* oligosaccharide repeat polymerase (WP_375217169.1).

The comparative analysis of EPS biosynthetic gene clusters between *P. toebii* H-70 and *P. thermoglucosidasius* DSM 2542$^T$, *P. thermantarcticus* DSM 9572$^T$, *P. galactosidasius* DSM 18751$^T$ was performed using BLASTp and visualized in Fig 4. The *P. galactosidasius* DSM 18751$^T$ EPS cluster exhibited the highest level of similarity to the *P. toebii*, with strong similarity and numerous high-identity matches (highlighted by red connections in the figure).

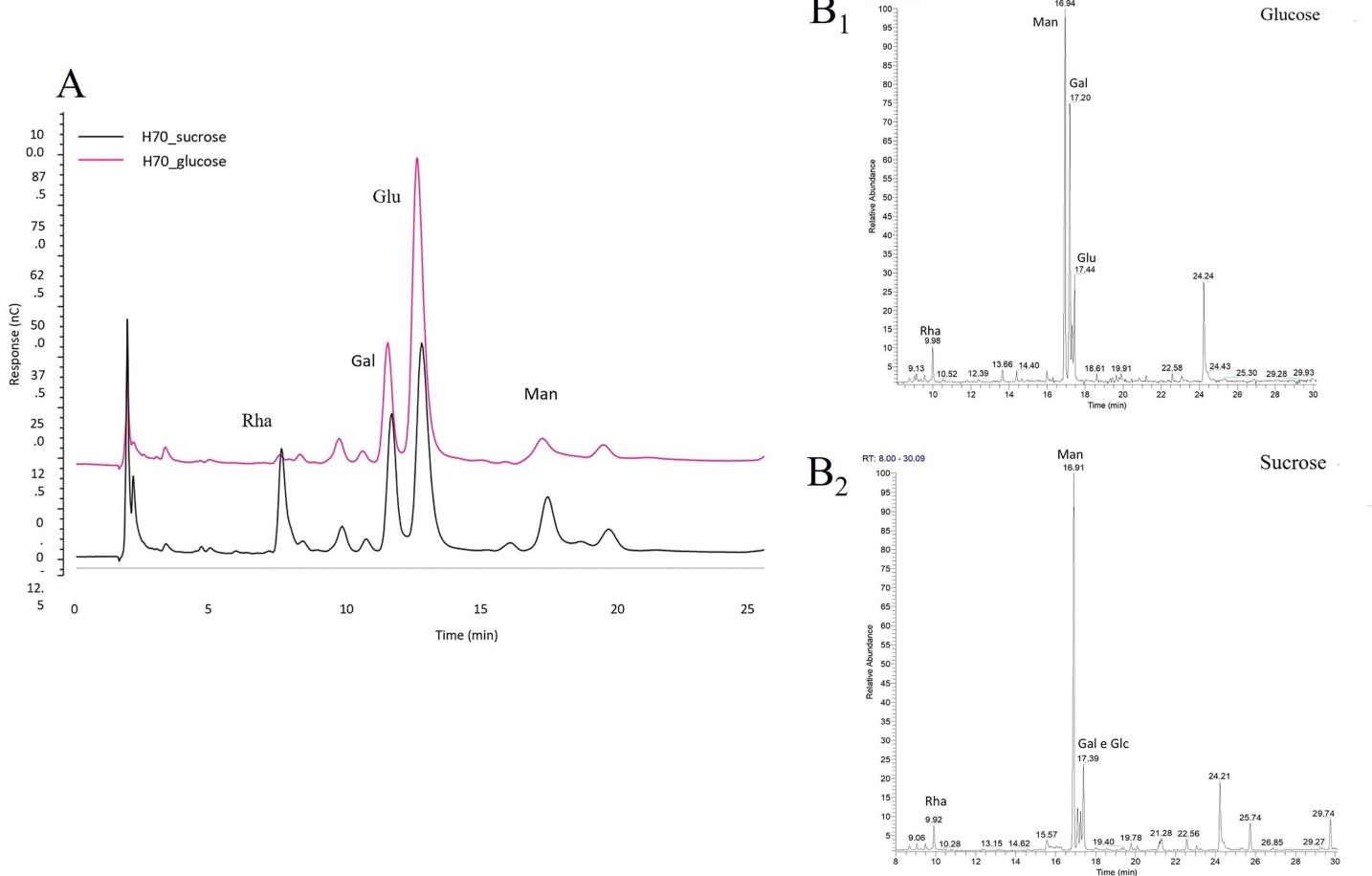

**Fig 1. EPS monomer composition: (A) HPAE-PAD analysis of the monosaccharide composition of EPS produced by *P. toebii* H-70 in presence of glucose (pink line) and sucrose (black line) as sole carbon source.** The chromatograms revealed the presence of rhamnose (RT: 7.8 min), galactose (Rt: 11.6 min), glucose (RT:12.9 min), and mannose (RT: 16.6 min). GC–MS analysis of methyl glyoside acetates of EPS produced by *P. toebii* H-70 in presence of glucose (B1) and sucrose (B2). Abbreviations: Rha, rhamnose; Man, mannose; Glu, glucose; Gal galactose.

**Table 1. Genome features and statistics of strain *P. toebii* H-70 strain.**

| GenBank accession | JAZHOX000000000 |
|---|---|
| No. contigs | 71 |
| No. base pairs | 3205758 |
| G+C content (mol %) | 42.0 |
| No. RNAs | 65 |
| No. genes (total) | 3509 |
| No. CRISPRs | 1 array, 3 spacers, 4 repeats |

## Carbohydrate active enzymes

The genome encodes a broad set of CAZymes, including auxiliary activities (AA), glycoside hydrolases (GH), glycosyl transferases (GT), carbohydrate esterases (CE), and carbohydrate-binding modules (CBM) families, as well as S-layer

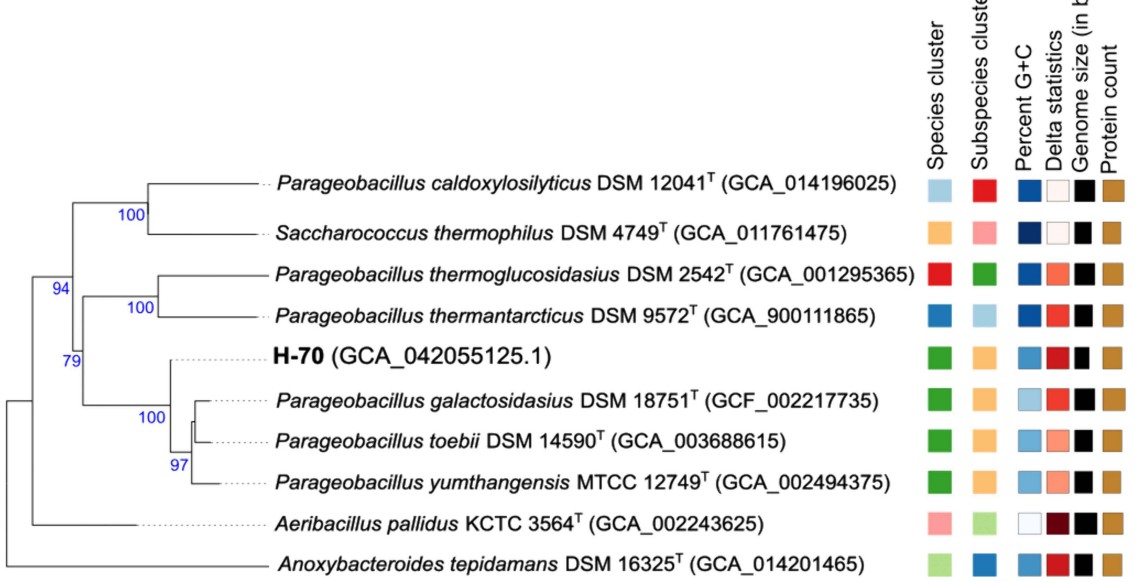

**Fig 2. Genome-based phylogenetic tree constructed using the GBDP method implemented in FastME version 2.1.6.1 [52].** Branch lengths were scaled according to GBDP distance formula d5. The numbers above branches indicate GBDP pseudo-bootstrap support values (> 60%) derived from 100 replications, with an average branch support of 90.8%. The tree was midpoint-rooted, and the δ statistic was 0.218.

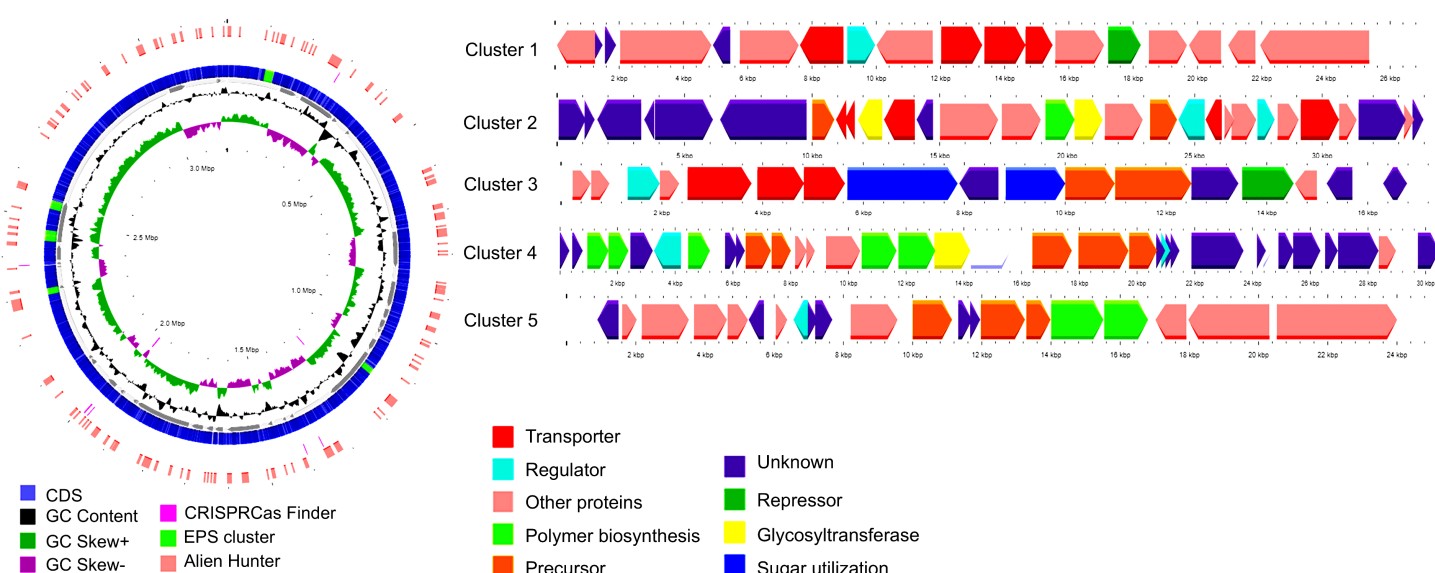

**Fig 3. The draft assembled circular genome of strain H-70, with the positions of EPS gene clusters highlighted and the structures of selected and extracted EPS biosynthesis gene clusters, with the scale shown in Kb.** From the inside to the outside, the features are arranged as follows: genome scale (in Mb), GC skew (+/−), GC content, contigs, CDSs with EPS clusters, CRISPR-Cas loci, and regions identified by Alien Hunter.

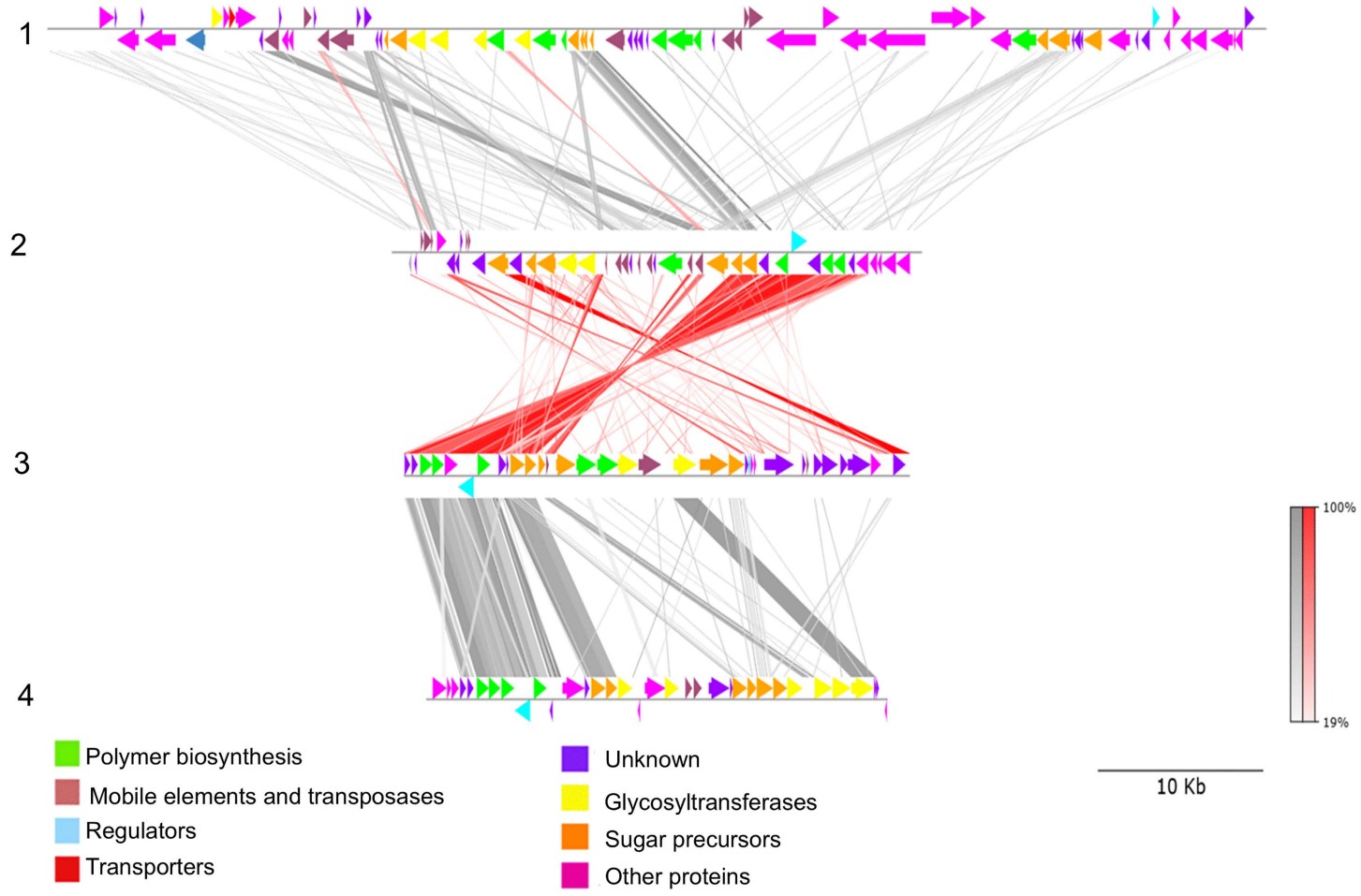

**Fig 4. The comparison of EPS gene clusters between the thermophilic bacteria *P. thermoglucosidasius* DSM2542ᵀ (1), *P. galactosidasius* DSM18751ᵀ (2), *P. toebii* H-70 (3) and *P. thermantarcticus* DSM 9572ᵀ (4) was done by pyGenomeViz (v1.0.0) genome visualization python package (Shimoyama, 2022) according to BLASTp (grey for matched in the same direction or red for inverted matches).** The genes in the clusters represent various colors to indicate their functional classifications. The scale is shown in kilobases (Kbp).

homology domains (SLH). AA4 and AA7 were the most abundant auxiliary activity families; CBM50 dominated CBM; CE4 was the major esterase family; GH13 was the most represented hydrolase; and GT2 was the most abundant transferase family. The distribution and abundance of AAs, CEs, GHs, GTs, CBMs, and SLH among *P. thermoglucosidasius* DSM 2542ᵀ, *P. thermantarcticus* DSM 9572ᵀ, *P. galactosidasius* DSM 18751ᵀ, and *P. toebii* H-70 are shown in Fig 5.

### Regulation of biosynthesis and EPS modification

EPS biosynthesis is regulated by multiple transcriptional regulators (e.g., LacI, AraC, LytR, MalR), two-component systems (DegU/DegS/DegV, CheA, ResE), and sporulation regulators (Spo0F/Spo0B/Spo0A). Several of these (AraC, LacI, DegU, DegV, LytR, MalR) were located within EPS clusters, indicating direct involvement. Genes encoding EPS modifications were also identified, including polysaccharide pyruvyl transferase (CsaB), probable deacetylase (pdaB), and diverse acetyltransferases (GNAT family and others). These enzymes likely contribute to structural variability and functional diversification of the EPS.

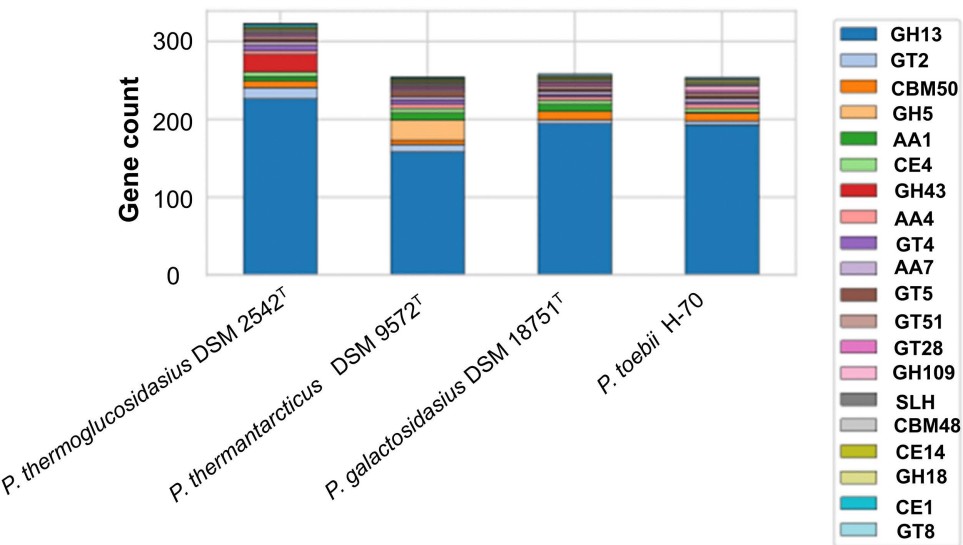

**Fig 5. The distribution and abundance of AAs, CEs, GHs, GTs, CBMs and SLH among *P. thermoglucosidasius* DSM 2542[T], *P. thermantarcticus* DSM 9572[T], *P. galactosidasius* DSM 18751[T] and *P. toebii* H-70.**

Genetic determinants of EPS biosynthesis pathways in *P. toebii* H-70 are detailed in S2 Table in S1 File.

The metabolic pathway included all necessary enzymes responsible for sugar transport into the cell until UDP sugar synthesis is demonstrated in Fig 6. Afterward, the mentioned proteins participating in the EPS assembly and export were shown in Fig 7.

### C-type lectins binding by ELISA-based solid-phase assay

The binding of *P. toebii* H-70 EPS to various human C-type lectins was analyzed using an ELISA-based solid-phase assay. Wells were coated with the EPS isolated from bacteria cultivated in either glucose- or sucrose-enriched medium. Subsequently, various C-type lectin-Fc chimeras were added, and binding was detected via an enzymatic colorimetric reaction (Fig 8). Consistent with the EPS's chemical structure and monosaccharide composition—which includes mannose, galactose, and rhamnose—strong binding to DC-SIGN, Langerin, and the mannose receptor (MR) was observed (S7 Fig in S1 File). In contrast, the macrophage galactose-type lectin (MGL) did not exhibit significant binding to the EPS. This is likely due to MGL's known specificity for terminal α-GalNAc residues, which are absent in the EPS structure. The different culture media (glucose or sucrose) did not significantly affect the lectin binding profile.

### Discussion

In this study, we provide the first comprehensive genomic and chemical characterization of EPS produced by the thermophilic bacterium *P. toebii* H-70, isolated from a geothermal spring in Armenia. The findings highlight both the distinctive structural features of the polymer and the genetic basis underlying its biosynthesis, expanding our understanding of EPS diversity among thermophilic bacilli. Thermophilic bacilli belonging to *Geobacillus*, *Parageobacillus*, *Bacillus*, and *Brevibacillus* have been reported as EPS producers from geothermal springs worldwide, including Italy, Bulgaria, and Turkey, in addition to polar region as Antarctica [19,22,25]. However, relatively little is known about EPS-producing microorganisms in Armenian geothermal ecosystems [24]. Recent studies described *Parageobacillus* isolates from the Arzakan geothermal spring in Armenia (*P. thermodenitrificans* ArzA-6 and *P. toebii* ArzA-8) as EPS producers [8]. Building on these

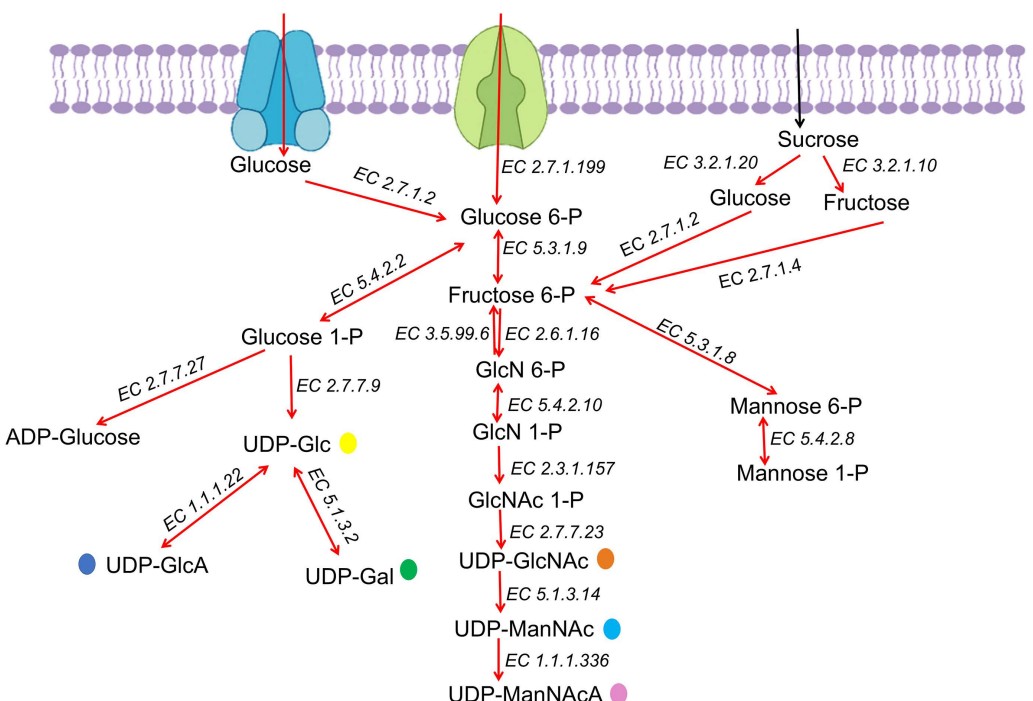

**Fig 6. Metabolic pathway map for EPS production in *P. toebii* H-70 strain illustrating the biosynthesis of activated nucleotide sugars based on genome functional annotation.** The enzymes are presented with EC numbers. **Abbreviations:** GlcN-6P: Glucosamine-6-phosphate, GlcN-1P: Glucosamine-1-phosphate, GlcNAc: N-Acetylglucosamine, UDP-Glc: Uridine diphosphate -glucose, UDP-GlcNAc: UDP N-acetylglucosamine, UDP-GlcA: UDP-D-glucuronic acid, UDP-Gal: UDP -galactose, UDP-ManNAc: UDP N-acetylmannosamine, UDP-ManNAcA: UDP-N-acetyl-D-mannosaminuronic acid, **Enzymes:** EC 5.1.3.2: UDP-glucose 4-epimerase; UDP-glucose dehydrogenase, EC 5.3.1.9: glucose-6-phosphate isomerase, EC 3.2.1.10: oligo-1,6-glucosidase, EC 2.7.1.2: glucokinase, EC 5.4.2.10: phosphoglucosamine mutase, EC 5.4.2.8: phosphomannomutase, EC 2.7.7.23: UDP-N-acetylglucosamine diphosphorylase, EC 5.1.3.14: UDP-N-acetylglucosamine 2-epimerase, EC 1.1.1.336: UDP-N-acetyl-D-mannosamine dehydrogenase, EC 5.3.1.8: mannose-6-phosphate isomerase: EC 2.7.1.199: D-glucose PTS permease, EC 5.4.2.2: phosphoglucomutase, EC 2.7.7.27: ADP-glucose synthase, EC 2.7.7.9: glucose-1-phosphate uridylyltransferase, EC 1.1.1.22: UDP-glucose dehydrogenase, EC 2.6.1.16: glucosamine-6-phosphate isomerase, EC 3.5.99.6: glucosamine-6-phosphate deaminase, EC 2.3.1.157: glucosamine-1-phosphate N-acetyltransferase, EC 2.7.1.4: fructokinase, EC 3.2.1.20: alpha-glucosidase.

observations, we screened over 100 thermophilic bacilli from Armenian geothermal springs and identified *P. toebii* H-70 from Hankavan as a promising EPS producer, distinguished by its mucous-like colonies on sucrose medium.

The EPS yield of H-70 is comparable to or slightly lower than values reported for other thermophilic bacilli, including *Parageobacillus* and *Anoxybacillus* spp. (Table 2), though differences in media composition and cultivation conditions complicate direct comparison. The EPS yield of H-70 (37.9 mg/L in sucrose medium, 0.10 g/g DCW)) was lower than that of some reported thermophilic bacilli, such as *P. toebii* ArzA-8 (80 mg/L) and *P. thermodenitrificans* B3-72 (70 mg/L) [8,53], but exceeded that of *Brevibacillus thermoruber* 423 and *Anoxybacillus kestanbolensis* 415 (28 and 25.3 mg/L, respectively) [54,55]. A *P. toebii* strain 419 from Bulgaria achieved 50 mg/L under optimized sucrose conditions [54]. Thus, while H-70's productivity is modest, it falls within the range of thermophilic producers and could be enhanced through optimization, by means of Random Surface Methodology (RSM), alternative cheaper growth media, and through fermentation parameter modulation [56]. EPS production in H-70 was growth-associated, with yields increasing alongside biomass from early growth through stationary phase. This agrees with prior reports that EPS synthesis in thermophiles is typically linked to active growth [54,55]. Although EPSs yields can vary as a function of the growth phase, most studies have shown that the EPSs levels remain constant throughout the batch cycle of growth [57,58]. Notably, the thermophilic process at 55 °C

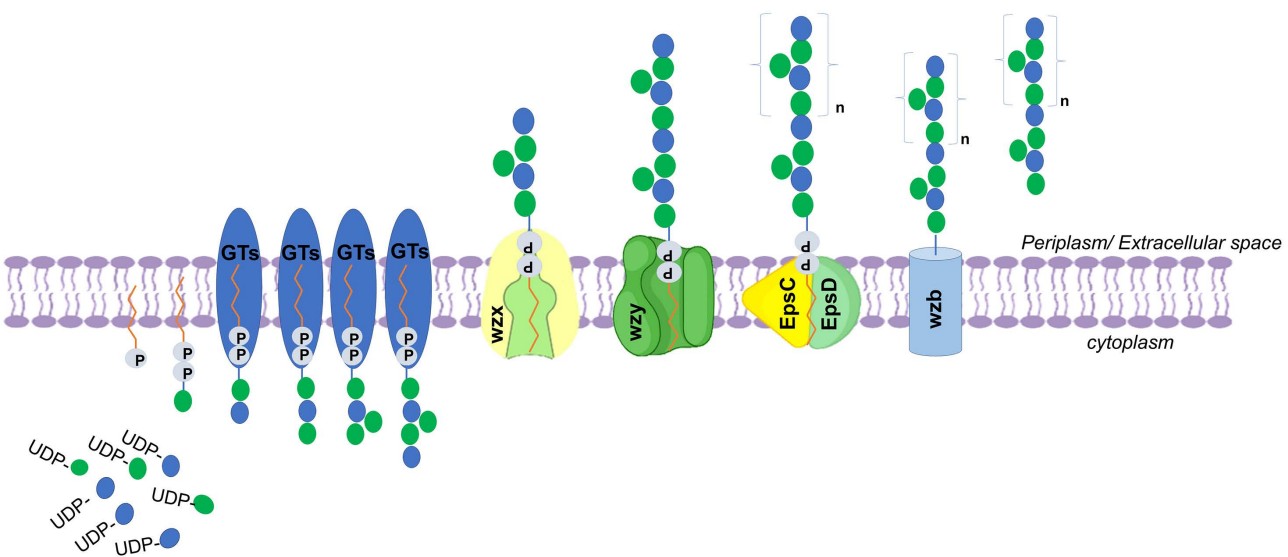

**Fig 7. General overview of the intracellular EPS biosynthesis pathway.** Display the Wzx/Wzy dependent pathway with the repeating unit assembled by various Glycosyltransferases (GT's) and the following translocation toward the periplasm by Wzx flippase. Polymerization occurs via Wzy polymerase. Chain length is determined by EpsC and EpsD proteins. The protein named wzb participates in the EPS synthesis and export.

enabled maximum EPS accumulation within a shorter cultivation period compared to mesophilic EPS systems, suggesting a biotechnological advantage in terms of process efficiency.

The EPS of H-70 was a heteropolysaccharide composed of rhamnose, glucose, galactose, and mannose, with additional uronic acids, proteins, and nucleic acids. Such chemical heterogeneity mirrors previous findings in extremophilic EPSs, where neutral and charged sugars combine with other extracellular polymers to form a multifunctional matrix [8,12]. These components are likely to enhance biofilm formation, metal binding, and emulsification capacity, broadening potential industrial applications [10].

Genome sequencing identified five EPS clusters in H-70, all consistent with Wzx/Wzy-dependent pathways. Comparative analysis with *P. thermoglucosidasius* DSM 2542[T], *P. galactosidasius* DSM 18751[T], and *P. thermantarcticus* DSM 9572[T] revealed strong conservation of structural genes (EpsC, EpsD, nucleotide-sugar precursor enzymes), highlighting the evolutionary stability of thermophilic EPS biosynthetic machinery (Fig 4). A gene annotated as hypothetical protein (Peg.1493) in H-70 showed 76.5% identity with EpsX in *P. thermantarcticus* DSM 9572[T], suggesting a role in polymer assembly and highlighting uncharacterized genes that may contribute to species-specific EPS features.

The CAZyme repertoire of H-70 was dominated by GH13 and GT2 families, consistent with a strong capacity for carbohydrate processing and polymerization. The presence of CBM50 modules and S-layer homology (SLH) domains suggests that EPS can be anchored to the cell wall, supporting biofilm formation. Comparisons with *P. galactosidasius* DSM 18751[T] revealed close similarity in GT2, CBM50, and SLH content, suggesting shared polysaccharide structures and conserved thermophilic adaptation strategies (Fig 5).

Regulatory analysis indicated the presence of Spo0A, AbrB, ClpC, DegU, LysR, MarR, and AraC family proteins. Spo0A-mediated repression of AbrB is a known switch for EPS operon activation [6], while AraC-type regulators have roles in biofilm formation in both Gram-positive and Gram-negative bacteria. Although LuxR-type proteins were identified, the absence of LuxI suggests an incomplete quorum sensing system, as also observed in *Geobacillus* sp. WSUCF1 [6]. Modification enzymes such as CsaB, acetyltransferases, and deacetylases were present, consistent with polymer tailoring through pyruvylation and acetylation. Such tailoring mechanisms may allow H-70 to fine-tune EPS function in response to

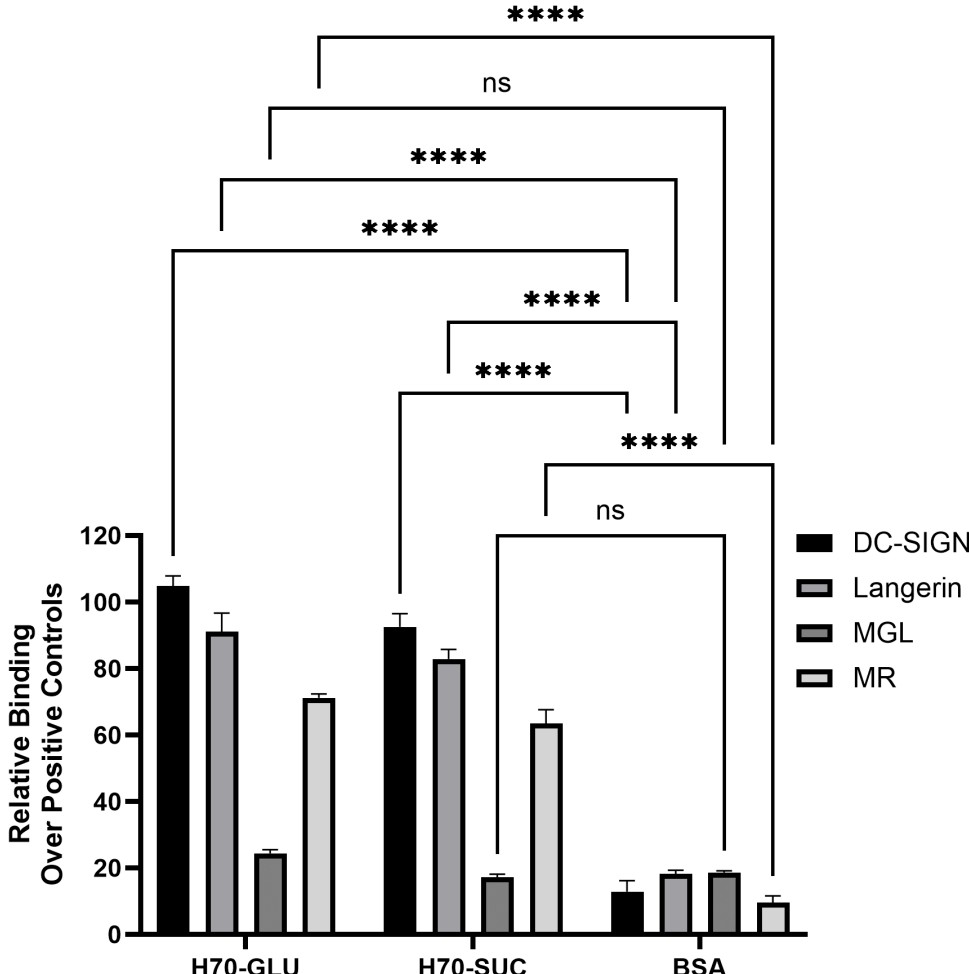

**Fig 8. Binding of EPS produced in glucose- or sucrose-enriched media was assessed for DC-SIGN, Langerin, MGL, and MR using a solid-phase assay in calcium-containing buffer.** Data points represent the mean of duplicate measurements, normalized to lectin-specific positive controls (glycan-functionalized polyacrylamide polymers or mannans). Error bars represent standard deviation. ns (not significant): $p > 0.05$, **** $p \leq 0.0001$. BSA was used as negative control.

environmental pressures. Polymer modifications can significantly influence the rheological properties and functional performance of the resulting exopolysaccharides. The degree of acetylation and pyruvylation exerts opposing effects on the viscosity of the resulting EPS. A high level of pyruvylation is associated with increased viscosity, while a greater degree of acetylation tends to reduce viscosity of the polymer [59].

The genomic repertoire supports a canonical Wzx/Wzy-dependent process: sugar units are activated as UDP-sugars, assembled into repeating units by glycosyltransferases, flipped across the cytoplasmic membrane by Wzx, polymerized by Wzy, and finally exported with chain-length control via EpsC/EpsD/EpsE proteins (Fig 7) [60,61]. Additional elements such as WaaL, previously characterized in Enterobacteriaceae for O-antigen ligation [62], were also detected, suggesting possible crosstalk between EPS biosynthesis and other cell-surface polysaccharides.

To explore the immunomodulatory potential of the EPS produced by *P. toebii* H-70, we investigated its interaction with key human C-type lectins. Recognition of carbohydrate-patterns is a fundamental immunological mechanism that mediates communication between bacteria and their hosts [63]. This process is facilitated by pattern recognition receptors,

**Table 2. Comparative table of EPS production and composition by thermophilic bacilli isolated from thermal sites.**

| Strain | Isolation source | Carbon source | EPS mg/L | Growth conditions | EPS composition | References |
|---|---|---|---|---|---|---|
| *P. toebii* H-70 | Hankavan hot spring, Armenia | Sucrose Glucose | 37.9 10.5 | Shake flask 55 ∘C, 72 h | Rha/Glc/Gal/Man | This study |
| *P. thermodenitrifcans* ArzA-6 | Arzakan geothermal spring, Armenia | fructose | 76 | Batch culture 65 °C, 20 h | Man/Gal/Ara/Fru/Glc (1/0.13/0.1/0.06/0.05) | [8] |
| *P. toebii* ArzA-8 | Arzakan geothermal spring, Armenia | fructose | 80 | Batch culture 65 °C, 20 h | Man/Gal/Glc/Ara (1/0.5/0.2/0.05) | [8] |
| *Geobacillus* sp. WSUCF1 | Compost, Pullman, WA | Glucose | 525.7 | Fermenter 60 ∘C, 24 h | EPS-1 Man/Glc (1:0.21) EPS-2 Man | [64] |
| *G. tepidamans* V264 | Velingrad hot spring, Bulgaria | Maltose | 111,4 | Fermenter 60 °C, 8 h | Glc/Gal/Fuc/Fru/0.07/0.04/0.02 | [18] |
| *B. thermoruber* 423 | Gradechnista hot spring, Bulgaria | Glucose Sucrose Maltose | 144 28 863 | Shake flask Shake flask Fermenter 55 °C, 24 h | Glc/Gal/Man/GalNAc/ManNAc (57.7/16.3/9.2/14.2/2.4) | [55] |
| *B. thermoruber* 438 | Rupi hot spring, Bulgaria | Maltose  Maltose | 29.41 78.1 | Shake flask Batch culture 55 °C, 8 h | NR | [19] |
| *Geobacillus (Parageoba-cillus) thermantarcticus* | Crater of Mount Melbourne, Antarctica | Mannose | 165 400 | Batch culture in bioreactor 65 °C, 24 h | EPSI, Man/Glc (1/0.7) | [16,65,66] |
| *Geobacillus* sp. 4004 | Flegrean hydrothermal vents, Ischia Island, Italy | Trehalose | 60 | Batch culture 75 °C, 48 h | EPS, Gal/Man/GlcN/ Ara (1.0/0.8/0.4/0.2) | [67,68] |
| *G. thermodenitrifcans* B3-72 | Shallow hydrothermal vent, Eolian Island, Italy | Glucose/ sucrose | 70 | Shake flask 60 °C, 72 h | EPS2, Man/Glc (1/0.2) | [69,70] |
| *G. stearothermophilus* 1A60 | Shallow hydrothermal vent, Panarea Island, Italy | Sucrose | <60 | Shake flask 50 °C, 72 h | NR, Man/Gal/GalN/Fuc/Glc (1:0.69:0.65:0.59:0.35) | [20] |
| *P. toebii* 419 | Rupi hot spring, Bulgaria | Sucrose | 50 | Shake flask 58 °C, 20 h | NR | [54] |
| *A. kenstanbolensis* 415 | Mizinka hot spring, Bulgaria | Sucrose | 25.3 | Shake flask 85 °C, 15 h | NR | [54] |
| *Anoxybacillus* sp. R4-33 | Radioactive radon hot spring, China | Glucose | 1083 | Shake flask 55 °C, 48 h | EPSII, Man/Glc (1/0.45) | [21] |
| *Aeribacillus pallidus* 418 | Rupi hot spring, Bulgaria | Maltose | 80 | Shake flask 55 °C, 20 h | EPSI, Man/Glc/ GalN/GlcN/Gal/Rib (1/0.16/0.09/0.07/0.06/0.04) EPSII, Man/Gal/Glc/ GalN/GlcN/Rib/Ara (1/0.52/0.45/0.34/0.23/0.15/0.14) | [54] |

Abbreviations: NR, not reported; Fru, Fructose; Fuc, Fucose; Gal, Galactose; Glc, Glucose; Man, Mannose; GalNAc, N-acetylgalactosamine; ManNAc, N-acetylmannosamine; GalN, Galactosamine; GlcN, Glucosamine; Rib, Ribose; Ara, Arabinose; Rha, Rhamnose.

including the conserved family of carbohydrate-binding proteins known as lectins. Our findings provide insight into how EPS from thermophilic bacteria interacts with immune lectins, which are expressed on various immune cells [32].

## Conclusions

This study provides the first comprehensive genomic and chemical characterization of extracellular polysaccharides produced by *P. toebii* strain H-70, a thermophilic bacterium isolated from an Armenian geothermal spring. Strain H-70 demonstrated strong

EPS productivity under sucrose-based cultivation, yielding a heteropolysaccharide composed of rhamnose, glucose, galactose, and mannose, along with proteins, uronic acids, and nucleic acids. Genomic analysis revealed a 3.2 Mb draft genome with high similarity to *P. toebii* DSM 14590$^T$ and identified five EPS biosynthetic clusters encoding Wzy-dependent pathway components, glycosyltransferases, transporters, and regulators. The presence of diverse CAZyme families and modification enzymes (e.g., CsaB, acetyltransferases) highlights the metabolic potential of H-70 to produce structurally versatile polysaccharides.

Together, these findings advance our understanding of thermophilic EPS biosynthesis, regulation, and structural diversity. The ability of *P. toebii* H-70 to synthesize complex EPSs under high-temperature conditions underscores its promise as a novel resource for biotechnological applications, particularly in industries requiring thermostable and functionally adaptable biopolymers. The binding of the EPS to different human C-type lectins, suggests a potential mechanism through which bacterial EPS could influence host immune responses. Future work should explore optimization of fermentation conditions, structural-functional relationships, and application-specific properties of H-70 EPSs to fully harness their industrial potential.

## Supporting information

**S1 File. Supplementary materials.**
(DOCX)

## Acknowledgments

The authors thank Dr. Salvatore Mallardo of IPCB-CNR for the thermogravimetrical analysis of EPS.

## Author contributions

**Conceptualization:** Armine Margaryan, Ilaria Finore, Annarita Poli, Hovik Panosyan.

**Data curation:** Diana Ghevondyan, Armine Margaryan, Yvette van Kooyk, Ilaria Finore, Annarita Poli, Hovik Panosyan.

**Formal analysis:** Diana Ghevondyan, Fabrizio Chiodo, Yvette van Kooyk, Ilaria Finore, Hovik Panosyan.

**Funding acquisition:** Annarita Poli, Hovik Panosyan.

**Investigation:** Armine Margaryan, Fabrizio Chiodo, Yvette van Kooyk, Annarita Poli, Hovik Panosyan.

**Methodology:** Diana Ghevondyan, Armine Margaryan, Fabrizio Chiodo, Yvette van Kooyk, Ilaria Finore, Hovik Panosyan.

**Project administration:** Annarita Poli, Hovik Panosyan.

**Resources:** Annarita Poli, Hovik Panosyan.

**Software:** Diana Ghevondyan, Armine Margaryan, Fabrizio Chiodo, Yvette van Kooyk, Ilaria Finore, Hovik Panosyan.

**Supervision:** Ilaria Finore, Annarita Poli, Hovik Panosyan.

**Validation:** Annarita Poli, Hovik Panosyan.

**Visualization:** Armine Margaryan, Fabrizio Chiodo, Yvette van Kooyk, Ilaria Finore, Annarita Poli, Hovik Panosyan.

**Writing – original draft:** Diana Ghevondyan, Armine Margaryan, Fabrizio Chiodo, Yvette van Kooyk, Ilaria Finore, Annarita Poli, Hovik Panosyan.

**Writing – review & editing:** Ilaria Finore, Annarita Poli, Hovik Panosyan.

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
