## [Decision Letter · Decision Letter 0]

11 Nov 2025

Dear Dr. Panosyan,

Thank you for submitting your manuscript to PLOS ONE. After careful consideration, we feel that it has merit but does not fully meet PLOS ONE’s publication criteria as it currently stands. Therefore, we invite you to submit a revised version of the manuscript that addresses the points raised during the review process.

We look forward to receiving your revised manuscript.

Kind regards,

Satheesh Sathianeson, Ph.D

Academic Editor

PLOS ONE

 [This research was funded by the Higher Education and Science Committee of MESCS RA within the framework of research projects N 23SC-CNR-1F010 and 23AA-1F043, by the European Commission – NextGenerationEU Project MICS (Made in Italy – Circular and Sustainable) n. PE00000004, CN_00000033 “National Biodiversity Future Center”- CUP B83C22002930006, PRIN MUR PNRR 2022 (P202293ZMC), FutuRaw FOE 2022 DCM.AD005.081, and MIMIT SANATEC F/350309/03/X60.]. 

Reviewers' comments:

Reviewer's Responses to Questions

**Comments to the Author**

1. Is the manuscript technically sound, and do the data support the conclusions?

Reviewer #1: Yes

Reviewer #2: Yes

2. Has the statistical analysis been performed appropriately and rigorously?

Reviewer #1: Yes

Reviewer #2: Yes

3. Have the authors made all data underlying the findings in their manuscript fully available?

Reviewer #1: Yes

Reviewer #2: Yes

4. Is the manuscript presented in an intelligible fashion and written in standard English?

Reviewer #1: Yes

Reviewer #2: Yes

Reviewer #1: Correct spacing errors (e.g., missing spaces between words, "kept" → "kept").

Use consistent units: mg/L⁻¹, µg/mL⁻¹, °C.

In figures, ensure all axes are labeled and units are given.

Table 2: Include references for each strain used in the comparison.

Use italics for scientific names (Parageobacillus toebii).

Avoid repetition between "Results" and "Discussion."

Reviewer #2: The work on Genomic and chemical insights into a human lectin-binding extracellular polysaccharides from Parageobacillus toebii strain H-70 is novel and contributing to the scientific community.

Minor comments:

- use uniform terms such as either CDS or CDSs.

- Add a reference for Geobacillus sp. WSUCF1

- Line-128: "sn" what is it?

- Can Authors add about thermal stability of the EPS, since it is from a thermophile?

- Share/show the binding data charts/graphs for human lectin binding experiments. How it is "significant/nonsignificant" as mentioned in line 303 and 305? Explain.

- Figure 3 need to be more contrast for visual understanding.

**Do you want your identity to be public for this peer review?** For information about this choice, including consent withdrawal, please see our Privacy Policy

Reviewer #1: No

Reviewer #2: **Yes: **

---

## [Author Response · Author response to Decision Letter 1]

18 Dec 2025

Author's Response to Decision Letter (PONE-D-25-54484

Genomic and chemical insights into a human lectin-binding extracellular polysaccharides from Parageobacillus toebii strain H¬-70)

Editor / Reviewer comments / Journal requirements in bold

Editor Comments:

Dear Dr. Panosyan,

Thank you for submitting your manuscript to PLOS ONE. After careful consideration, we feel that it has merit but does not fully meet PLOS ONE’s publication criteria as it currently stands. Therefore, we invite you to submit a revised version of the manuscript that addresses the points raised during the review process.

Author's Response: We sincerely appreciate the time and effort you have dedicated to reviewing our manuscript titled "Genomic and chemical insights into a human lectin-binding extracellular polysaccharides from Parageobacillus toebii strain H¬-70)". We have carefully considered all your comments and suggestions, and we have addressed them in detail in the revised manuscript.

The point-by-point response to the comments of Editor and reviewers have been uploaded as a separate file labeled Response to Reviewers. All the amends on our manuscript are highlighted in yellow. A marked-up copy of our manuscript that highlights changes made to the original version have been uploaded as a separate file labeled 'Revised Manuscript with Changes'. An unmarked version of our revised paper without highlighted changes have been upload as a separate file labeled 'Manuscript'.

Author's Response: We have carefully revised the manuscript to ensure full compliance with PLOS ONE style guidelines, including formatting, structure, and file naming conventions. The manuscript has been prepared using the official PLOS ONE formatting templates for the main text and the title/authors/affiliations pages, as recommended.

Author's Response: The fieldwork and sampling activities were conducted within the framework of a state-funded research project, which officially authorized access to geothermal spring sites and permitted the isolation of microbial samples. The full name of the approving authority and details of the authorization have now been explicitly added to the Methods section. As the work was performed under this governmental approval, no separate individual permits were required.

Please check in revised version of the manuscript:

Page 6, Lines 125-129

‘’Field sampling and isolation of microbial strains from geothermal springs were conducted within the framework of a state-funded research project officially approved by the Ministry of Education, Science, Culture and Sports of the Republic of Armenia, which authorized access to the field sites and permitted the collection of environmental samples. No additional individual permits were required’’.

[This research was funded by the Higher Education and Science Committee of MESCS RA within the framework of research projects N 23SC-CNR-1F010 and 23AA-1F043, by the European Commission – NextGenerationEU Project MICS (Made in Italy – Circular and Sustainable) n. PE00000004, CN_00000033 “National Biodiversity Future Center”- CUP B83C22002930006, PRIN MUR PNRR 2022 (P202293ZMC), FutuRaw FOE 2022 DCM.AD005.081, and MIMIT SANATEC F/350309/03/X60.].

Author's Response: We confirm that funders had no role in study design, data collection and analysis, decision to publish, or preparation of the manuscript. The Role of Funder statement has been added in both revised manuscript and cover letter.

Please check in revised version of manuscript:

Page 19, Lines 454-455

‘’The funders had no role in study design, data collection and analysis, decision to publish, or preparation of the manuscript’’.

Author's Response: We confirm that, upon careful review of all reviewer comments, none of the reviewers recommended or requested the inclusion of specific previously published works. Accordingly, no additional references were added, as there was no indication from either the reviewers or the Editor that such citations were required.

Author's Response: We have carefully reviewed the entire reference list to ensure that it is complete, accurate, and up to date. All cited references were checked for correctness, including author names, titles, journal information, and DOIs where applicable. We confirm that none of the cited articles have been retracted. Therefore, no retracted publications are included in the revised manuscript. As a result, no references required removal or replacement on this basis. No changes to the reference list were necessary beyond minor formatting corrections, which have been implemented in accordance with PLOS ONE guidelines.

Reviewers' comments:

Reviewer #1:

Reviewer: Correct spacing errors (e.g., missing spaces between words, "kept" → "kept").

Author's Response: We sincerely appreciate the time and effort you have dedicated to reviewing our manuscript. Thank you for pointing this out. All spacing errors and minor typographical issues (including missing spaces between words) have been carefully reviewed and corrected throughout the manuscript.

Reviewer: Use consistent units: mg/L⁻¹, µg/mL⁻¹, °C.

Author's Response: Thank you for this comment. We have revised the manuscript to ensure consistent use of units throughout the text. All concentration units are now uniformly presented as mg/L or µg/mL, and all temperature values are consistently reported in °C.

Reviewer: In figures, ensure all axes are labeled and units are given.

Author's Response: Thank you for this comment. All figures have been carefully revised to ensure that all axes are clearly labeled and that the appropriate units are provided where applicable.

Reviewer: Table 2: Include references for each strain used in the comparison.

Author's Response: We have carefully addressed your comment and have now included references for each strain used in the comparison throughout the manuscript. All relevant citations have been added in the text and listed in the reference section.

Reviewer: Use italics for scientific names (Parageobacillus toebii).

Author's Response: We confirm that all requested changes have been made. Specifically, all scientific names, including Parageobacillus toebii, have been formatted in italics throughout the manuscript.

Reviewer: Avoid repetition between "Results" and "Discussion."

Author's Response: We thank the editor for the valuable suggestion. We have carefully revised the manuscript to minimize repetition between the Results and Discussion sections. The Results section now focuses strictly on presenting the findings, while the Discussion section has been refined to interpret and contextualize these results.

Reviewer #2:

Reviewer: The work on Genomic and chemical insights into a human lectin-binding extracellular polysaccharides from Parageobacillus toebii strain H-70 is novel and contributing to the scientific community.

Minor comments:

use uniform terms such as either CDS or CDSs.

Author's Response: We sincerely thank you for your careful review and valuable suggestions. We have carefully revised the manuscript in accordance with your comments. Specifically, we have ensured that terminology regarding coding sequences is now used consistently throughout the text. All instances have been standardized to “CDSs”.

Reviewer: Add a reference for Geobacillus sp. WSUCF1

Author's Response: A reference for Geobacillus sp. WSUCF1 has been added as suggested.

Reviewer: Line-128: "sn" what is it?

Author's Response: The issue on Line 128, where “sn” appeared, was a typographical error. This has now been removed in the revised manuscript.

Reviewer: Can Authors add about thermal stability of the EPS, since it is from a thermophile?

Author's Response: Thank you very much for this comment. In order to verify the thermal stability of the EPS, we have performed thermogravimetric analysis, and the results are now included in the revised manuscript (please see Supplementary Figure S5).

Please check in revised version of the manuscript:

Pages 7-8, Lines 166-169

‘’Thermogravimetrical analysis (TGA) of EPS was performed with TGA/DTA Perkin-Elmer PyrisDiamond, with gas station. Approximately 4 mg of sample (kept under vacuum at 60 °C for 24 h prior analysis) were placed in a ceramic crucible and heated from room temperature up to 600 °C at a speed rate of 10 °C/min, under nitrogen flux at 30 mL/min’’.

Page 11, Lines 256-266

‘’Determination of the thermal behavior of EPS is important in order to evaluate its commercial application in packaging or other industrial use. Thermogravimetric analysis is a thermal analysis in which the mass of the sample is measured over time as the temperature changes. It can be used to evaluate the thermal stability of the material. S5 Fig shows TGA of EPS. Firstly, a step around 80-100 °C was observed that could be related to water evaporation from the EPS. This behavior is due to the initial loss of moisture related to the presence of high content of carboxyl groups, which are bound to water molecules [4]. A second step, related to the degradation of bigger part of the sample, was centered between 200 and 350 °C (at 300 °C, more than 50% of EPS was still present). According to thermal characteristics of EPS which are comparable to those from commercial biopolymers, this polymer could be considered as thermostable and could be therefore used in thermal processed below 200 °C without significant thermal degradation risk’’.

Please check in revised version of the Supplementary materials:

S5 Fig. Thermogravimetrical analysis of EPS from strain H-70 grown in sucrose containing medium.

Reviewer: Share/show the binding data charts/graphs for human lectin binding experiments. How it is "significant/nonsignificant" as mentioned in line 303 and 305? Explain.

Author's Response: We have added the requested statistical analysis to clarify the significance of our results. The updated Figure 8 now includes statistical comparisons for human lectin binding experiments. Additionally, in the Supporting Material, we have provided the raw lectin-binding data (Optical Density at 450 nm) so that readers can directly assess the binding levels (S7_Fig).

Please check in revised version of the manuscript:

Page 32, 736-741

‘’Fig 8. Binding of EPS produced in glucose- or sucrose-enriched media was assessed for DC-SIGN, Langerin, MGL, and MR using a solid-phase assay in calcium-containing buffer. Data points represent the mean of duplicate measurements, normalized to lectin-specific positive controls (glycan-functionalized polyacrylamide polymers or Mannans). Error bars represent standard deviation. ns (not significant): p> 0.05, **** p ≤ 0.0001. BSA was used as negative control’’.

Please check in revised version of the Supplementary materials:

S7 Fig. Binding of EPS produced in glucose- or sucrose-enriched media was assessed for DC-SIGN, Langerin, MGL, and MR using a solid-phase assay in calcium-containing buffer. Data points represent the mean of duplicate measurements measured as optical density (OD) at λ 450nm. Error bars represent standard deviation.

Reviewer: Figure 3 need to be more contrast for visual understanding.

Author's Response: We have carefully revised Figure 3 to increase its contrast. The updated figure has been uploaded in the revised manuscript.

---

## [Editor Report · Decision Letter 1]

21 Dec 2025

Genomic and chemical insights into a human lectin-binding extracellular polysaccharides from Parageobacillus toebii strain H-70

PONE-D-25-54484R1

Dear Dr. Panosyan,

We’re pleased to inform you that your manuscript has been judged scientifically suitable for publication and will be formally accepted for publication once it meets all outstanding technical requirements.

Kind regards,

Satheesh Sathianeson, Ph.D

Academic Editor

PLOS One
---

## [Editor Report · Acceptance letter]

PONE-D-25-54484R1

PLOS One

Dear Dr. Panosyan,

I'm pleased to inform you that your manuscript has been deemed suitable for publication in PLOS One. Congratulations! Your manuscript is now being handed over to our production team.

Kind regards,

on behalf of

Dr. Satheesh Sathianeson

Academic Editor

PLOS One